**Data Availability Statement:** Due to French law there are restrictions on publicly sharing the data of this study. French law requires that everyone who

---

# Intimate partner violence by men living with HIV in Cameroon: Prevalence, associated factors and implications for HIV transmission risk (ANRS-12288 EVOLCAM)

**Marion Fiorentino**[1]*, **Abdourahmane Sow**[1], **Luis Sagaon-Teyssier**[1], **Marion Mora**[1], **Marie-Thérèse Mengue**[2], **Laurent Vidal**[1], **Christopher Kuaban**[3], **Laura March**[4], **Christian Laurent**[5], **Bruno Spire**[1], **Sylvie Boyer**[1], for the EVOLCam study Group¶

1 INSERM, IRD, Aix Marseille Univ, SESSTIM, Sciences Economiques & Sociales de la Santé & Traitement de l'Information Médicale, Marseille, France, 2 Université Catholique d'Afrique Centrale, Yaoundé, Cameroun, 3 Department of Internal Medicine and Subspecialties, Faculty of Medicine and Biomedical Sciences, University of Yaoundé 1, Yaoundé, Cameroon, 4 Laboratoire Populations Environnement Développement, UMR 151, IRD, Aix-Marseille University, Marseille, France, 5 IRD UMI 233-INSERM U1175, Montpellier Univ, Montpellier, France

¶ Membership of the EVOLCam study group is provided in the Acknowledgments.
* marion.fiorentino@inserm.fr

## Abstract

### Objectives

Intimate partner violence (IPV) against women is frequent in Central Africa and may be a HIV infection risk factor. More data on HIV-positive men (MLHIV) committing IPV are needed to develop perpetrator-focused IPV and HIV prevention interventions. We investigated the relationship between IPV and HIV transmission risk and IPV-associated factors.

### Methods

We used data from the cross-sectional survey EVOLCam which was conducted in Cameroonian outpatient HIV structures in 2014. The study population comprised MLHIV declaring at least one sexual partner in the previous year. Using principal component analysis, we built three variables measuring, respectively, self-reported MLHIV-perpetrated psychological and physical IPV (PPV), severe physical IPV (SPV), and sexual IPV (SV). Ordinal logistic regressions helped investigate: i) the relationship between HIV transmission risk (defined as unstable aviremia and inconsistent condom use) and IPV variables, ii) factors associated with each IPV variable.

### Results

PPV, SPV and SV were self-reported by 28, 15 and 11% of the 406 study participants, respectively. IPV perpetrators had a significantly higher risk of transmitting HIV than non-IPV perpetrators. Factors independently associated with IPV variables were: i) socio-demographic, economic and dyadic factors, including younger age (PPV and SPV), lower income

---

wishes to access cohorts data or clinical study data on humans must ask the French data protection authority, la Commission Nationale de l'Informatique et des Libertés (CNIL), for permission by filling a form which can be provided by Gwenaëlle Maradan (Observatoire Régional de la Santé PACA ie. The Regional Health Observatory PACA mail: gwenaelle.maradan@inserm.fr). For further information, please see: https://www.cnil.fr/ .

**Funding:** This study was funded by the French National Agency for Research on AIDS and viral hepatitis (ANRS) (website anrs.fr/en) The funder had no role in study design, data collection and analysis, decision to publish, or preparation of the manuscript.

**Competing interests:** No authors have competing interests.

(PPV), not being the household head (SPV and SV), living with a main partner (SPV), and having a younger main partner (SPV); ii) sexual behaviors, including ≥2 partners in the previous year (PPV and SPV), lifetime sex with another man (SPV), inconsistent condom use (SV), and >20 partners during lifetime (SV); iii) HIV-related stigma (PPV and SV).

## Conclusion

IPV perpetrators had a higher risk of transmitting HIV and having lifetime and recent risky sexual behaviors. Perpetrating IPV was more frequent in those with socioeconomic vulnerability and self-perceived HIV-related stigma. These findings highlight the need for interventions to prevent IPV by MLHIV and related HIV transmission to their(s) partner(s).

## Introduction

Intimate partner violence (IPV) affects two thirds of the population in Central Africa [1]. Growing evidence suggests that, in certain contexts, IPV—including non-sexual IPV—increases the risk of HIV acquisition in women in heterosexual relationships through three main mechanisms [2–4]. First, men who are violent with female partners are more likely to acquire HIV from other partners, because of more frequent risky behaviors, including unprotected sex, transactional sex, and higher sexually transmitted infection (STI) incidences [5–9]. Second, women may have greater difficulty negotiating condom use or refusing to have sex with their partners [9]. Coerced or undesired sex may also lead to more frequent anal intercourse [10] and anatomic lesions of the vaginal or rectal mucosa which increase the risk of HIV acquisition [7]. Third, the psychological consequences of IPV on a victim may deteriorate immunity [5, 11], which is also a higher risk factor for HIV acquisition.

In addition, being HIV infected may amplify existing psychological and social difficulties or generate new ones [6, 12–16], as the trauma of HIV diagnosis is compounded for those with prior traumatic stressors [17, 18].

Few studies to date have examined the characteristics of men living with HIV (MLHIV) who perpetrate IPV other than sexual behaviors [19]. Furthermore, the role which HIV infection-specific factors play in IPV (such as MLHIV diagnosis following partner's HIV positive status disclosure, perception of HIV-related stigma, and experience with antiretroviral therapy (ART) including adherence and treatment interruption), has not been explored in this population. The profiles of IPV perpetrators are diverse, as are the psychosocial mechanisms underlying IPV [20]. Studies have highlighted the need for further research in this area to improve IPV prevention interventions [21]. A better understanding of the characteristics of MLHIV who commit IPV could help to identify potential perpetrators, while understanding the factors underlying the relationship between IPV and HIV transmission risk could help prevent transmission to their partners.

The HIV epidemic is generalized in Cameroon, with an overall prevalence of 4% in adults aged 15–49 years and 5% in women, which is twice as high as in men [22]. Between 12 and 22% of people living with HIV (PLHIV) are in a stable HIV-serodiscordant relationship [23, 24]. One third of PLVIH receiving ART in the national ART access program are not virally suppressed [25, 26]. Furthermore, according to the 2011 Demographic Health Survey, IPV is widespread in the general population (HIV positive or not), with half of women and men reporting IPV victimization and perpetration, respectively [23]. Although a small number of studies have investigated the role of individual factors in HIV transmission risk in Cameroon

[27–30], neither the specific role of IPV in HIV transmission risk nor the factors associated with MLHIV-perpetrated IPV have ever been explored there. To our knowledge, the same is true for all the other countries in Central Africa.

Our primary study hypothesis was the following: MLHIV perpetrating IPV have a higher risk of transmitting HIV because of frequent risky sexual behaviors and/or a higher likelihood of having a detectable viral load. We also hypothesized that living with HIV might generate specific psychosocial effects which increase the risk of MLHIV perpetrating IPV. Accordingly, this study aimed to: i) describe the various forms of self-reported MLHIV-perpetrated IPV; ii) investigate whether perpetrators had a greater risk of transmitting HIV to their female partners than non-perpetrators (intermediary analysis); iii) identify the demographic, socioeconomic, dyadic, psychosocial and behavioral characteristics of MLHIV perpetrators of IPV (main analysis).

## Methods

### Study design and data collection

We used data from the EVOLCam (ANRS-12288) cross-sectional survey, which aimed to study the living conditions of PLHIV linked to care in Cameroon's national ART access program, specifically in the Littoral and Center regions. Eligible patients (≥21 years old and diagnosed with HIV ≥3 months) attending one of the survey's 19 participating HIV services between April and December 2014, were randomly selected and invited to participate. The study protocol and participant inclusion procedures are described in detail elsewhere [31].

Briefly, eligible patients were invited to participate in EVOLCam during a HIV follow-up consultation. Those who agreed were referred to a trained independent interviewer to answer a face-to-face questionnaire which gathered demographic, socioeconomic, behavioral, psychosocial, domestic and dyadic data. Male participants answered a specific 'perpetuated IPV' section, comprising 12 behavior-specific questions about any acts of violence they might have perpetrated against their most recent female partner in the 12 months prior to the survey, and the frequency of these acts (frequent, occasional, never). The questions were adapted from the World Health Organization's (WHO) IPV victimization questionnaire for women [32], which has been used in previous studies to assess IPV perpetration [8]. Information about sexual behaviors, HIV status disclosure to a partner and partner's HIV status was also gathered for their most recent female partner (or two most recent female partners for those who reported more than one) in the 12 months prior to the survey. The questionnaire modules corresponding to data presented in this article are included in S1 and S2 Files. Unless otherwise specified, the term 'partner(s)' in this article refers to female partner(s).

After the interviews, blood samples were collected and analyzed in a reference laboratory in Yaoundé to measure viral load (only for ART-treated patients >6 months) and CD4 cell count measurements. The quantification threshold for viral load measurement was 100 copies/mL. Clinical data were collected from both medical files and clinical examinations using standardized medical questionnaires.

Before the study's implementation, a pilot survey was conducted to test the questionnaires and data collection procedures in six urban and rural hospitals. All participants provided written informed consent. The ANRS-12288 EVOLCam study was approved by the Ministry of Public Health in Cameroon and the Cameroonian National Ethics Committee.

### Study population

The study population included MLHIV who declared at least one female partner in the 12 months prior to the survey and who had no missing data for the study questionnaire's 12 'perpetrated IPV' items.

## Study outcomes

**Main outcomes.** The three main study outcomes were different forms of IPV perpetrated by MLHIV (psychological and physical IPV (PPV), severe physical IPV (SPV), and sexual IPV (SV)). All three were created using the methods described in the subsection "Statistical analysis" (see below) and each was defined as a three-class categorical variable reflecting the level of violence perpetrated (e.g., no PPV; moderate level of PPV; high level of PPV).

**Secondary outcomes.** The secondary study outcomes were the two following binary variables: 'unstable aviremia' and 'HIV transmission risk'. Participants were classified with unstable aviremia if i) they were not currently on ART or were on ART for less than 6 months; ii) if they were on ART for at least 6 months but had a detectable viral load and/or were poorly adherent to ART (defined as taking <80% of their prescribed ART doses and/or reporting treatment interruptions for at least two consecutive days in the 4 weeks prior to the survey [27]). Given that viral suppression prevents HIV transmission to HIV-negative partners in heterosexual couples [33], HIV transmission risk was defined as a combination of unstable aviremia and inconsistent condom use with their most recent or, for those who reported more than one partner in the previous 12 months, with at least one of their two most recent female partners of negative or unknown HIV status. Inconsistent condom use was defined as replying "Never, sometimes or almost always" to the survey question "In the last 12 months, have you used condoms with this partner?", and/or replying "No" to the question "During your most recent sexual intercourse with this partner, did you use a condom?".

## Explanatory variables

The following variables were considered in the analysis:

- demographic and socioeconomic characteristics: age, residential setting (urban vs. rural), educational level, occupational category, household monthly income;

- domestic and dyadic characteristics: being the household head, number of children, having a main female partner (if reported) in the previous 12 months, living with main partner at time of survey, type of union (marriage or common-law union), polygamous or monogamous union, relationship duration, already in a relationship with current main partner at the time of respondent's HIV diagnosis, currently desiring or trying to have a child with partner, educational level disparity with main partner, age disparity with main partner, main partner involvement in decision-making about how to spend respondent's income, main partner involvement in decision-making about respondent's healthcare, being in a serodiscordant couple, breaking up with a main partner in the 12 months prior to the survey because of HIV;

- sexual behaviors in the 12 months prior to the survey: number of partners, inconsistent condom use with the most recent or—if more than one partner declared in that timeframe—at least one of the two most recent partners, HIV status of the most recent or at least one of the two most recent partners, HIV-positive status disclosure to the most recent or at least one of the two most recent partners, transactional sex (paid or received);

- lifetime sexual behaviors: number of partners, sex with another man;

- clinical characteristics and experience of living with HIV: time since HIV diagnosis, HIV diagnosis following partner's diagnosis, time between diagnosis and ART initiation, self-reported adherence to ART in the 4 weeks prior to the survey (high adherence; low adherence; not receiving ART [34]), ART interruption >1 month since treatment initiation, knowledge about benefits of ART vis-à-vis prevention of sexual and mother-to-child

transmission, respondent's perception of HIV-related stigma (score 0–8 computed using the HIV Stigma Scale [35]);

- other psychosocial variables: mental quality of life (measured using the SF12 scale [36]) and frequent binge drinking (defined as drinking ≥3 large bottles of beer (i.e., ≥260 cL in total) and/or 6 other alcohol on one occasion at least once a month).

## Statistical analysis

**Construction of the IPV outcomes.** A principal component analysis (PCA) of the 12 acts of violence and their frequency (never, sometimes, often), corresponding to the 12 items in the 'perpetrated IPV' questionnaire section was conducted. The Promax rotation technique, which takes into account correlation between factors, was implemented to improve the fit of the data [37]. The PCA resulted in three scores of IPV, which were standardized and computed using values ranging from 0 to 1. These three scores corresponded to three forms of IPV which we defined as follows (S1 Table): i) psychological and physical IPV (PPV), which included 5 items (humiliation, insults or belittlement, threats, shoving or pushing or object throwing, slapping; eigenvalue = 3.2, Cronbach's α = 0.83); ii) severe physical IPV (SPV), which included 5 items (shoving or pushing or object throwing, slapping, arm twisting or hair pulling, punching or hitting, kicking or dragging or beating up; eigenvalue = 2.3, Cronbach's α = 0.69); iii) sexual IPV (SV) which included 2 items (forced sexual intercourse and any forced sexual act; eigenvalue = 1.8, Cronbach's α = 0.62). No respondent reported choking or burning a partner, or using or threatening to use a gun, knife or other weapon against a partner. The standardized scores were considered very reliable and reliable, respectively, when the Cronbach α value was ≥0.7 and [0.5; 0.7] [38, 39]. The three IPV scores explained 73% of the cumulative variance.

For each standardized IPV score a three-class variable (corresponding to our three study outcomes) reflecting the level of violence perpetrated was built using the following individual score cut-offs: score = 0 (no PPV, SPV or SV, as relevant), score <median among non-zero values (moderate level, as relevant) and score ≥median among non-zero values (high level, as relevant).

**Intermediary analysis: Relationship between HIV transmission risk and different outcomes.** The proportions of MLHV with unstable aviremia and HIV transmission risk were described overall and according to each IPV outcome (PPV, SPV, PV). With regard to the latter, we used univariate ordinal logistic regressions to test for significant differences between the proportions for the three outcomes.

**Main analysis: Factors associated with each outcome.** Univariate ordinal logistic regressions were performed to investigate the associations between each outcome and the explanatory variables listed above. Covariates with a p-value <0.2 in the univariate analysis were considered eligible for the multivariate ordinal logistic regression models. A backward selection method was used to select the covariates for the final multivariate model with a p-value <0.05. All statistical analyses were performed using SAS 9.4 and RSTUDIO 1.1.453.

## Results

### Study population characteristics

Of the 2138 participants in the ANRS-12288 EVOLCam survey, 572 (27%) were male. Among them, 446 (78%) declared at least one partner in the 12 months prior to the survey. Of the latter, 40 (9%) were excluded because of incomplete data (i.e., at least one of the 12 items in the 'IPV perpetration' section of the questionnaire was missing). The study population therefore included 406 MLHIV.

Median age was 43 [interquartile range (IQR): 27–51] years (Table 1). Most MLHIV were living in urban areas (83%), were professionally active (79%) and the median [IQR] monthly household income was 19 USD [IQR: 9–36] per adult-equivalent. A large majority had a main partner (88%). Three-quarters (76%) had a partner at least 5 years younger than themselves, and half (53%) were in a serodiscordant couple. One third reported at least two sexual partners (main or casual) in the 12 months prior to the survey, and half reported inconsistent condom use with their most recent or at least one of their two (if more than one partner declared) most recent partners. One quarter of MLHIV experienced HIV-related stigma and reported frequent binge drinking. Median time since HIV diagnosis was 3 years 9 months [IQR: 1 years 5 months– 6 years 10 months]. Eighty-eight percent of the study population were on ART at the time of the survey and among them, 29% reported high adherence.

## Descriptive analysis of IPV variables

Twenty nine percent of participants self-reported perpetrating PPV (14% moderate level, 14% high level), 15% SPV (7%, 8%), and 11% SV (6%, 5%) (Table 1).

## Relationship between HIV transmission risk and IPV perpetration

Fig 1 describes unstable aviremia and HIV transmission risk variables, both overall and according to the three IPV outcomes. Overall, 84% of MLHIV had unstable aviremia, while 27% were at risk of transmitting HIV. Fig 1 shows that the proportion of those with unstable aviremia was significantly higher in moderate level PPV perpetrators than in respondents not reporting PPV (non-perpetrators hereafter) (98% vs. 82%, p = 0.014). The proportions of unstable aviremia according to the SVP and SV variables were also higher in moderate level and high level perpetrators than in non-perpetrators but these differences were not significant at the 5% level. Additionally, the proportions of those at risk of transmitting HIV were significantly higher in moderate level perpetrators of PPV, SPV and SV than in non-perpetrators (45% vs. 24%, p<0.001; 44% vs. 25%, p = 0.031; 46% vs. 26%, p = 0.036, respectively). However, no significant difference in the proportions of HIV transmission risk was found in high level perpetrators of PPV, SPV and SV compared with non-perpetrators.

## Factors associated with IPV perpetration

Factors eligible for multivariate models are presented in Table 2 and results of the final multivariate models are showed in Fig 2. Three main groups of variables were significantly associated with IPV perpetration: i) socio-demographic, economic and dyadic factors ii) sexual behaviors iii) HIV-related stigma. More precisely, multivariate analysis showed that younger participants (adjusted Odds-Ratio (aOR) confidence interval (CI): 0.98 95% [0.96–0.99] per year), those with lower monthly household income (0.98 [0.96–0.99], per 1000 FCFA i.e., 1.66 USD per adult-equivalent), those perceiving HIV-related stigma (1.15 [1.03–1.27]) and those reporting more than one partner in the 12 months prior to the survey (1.84 [1.16–2.91]) were all significantly more likely to perpetrate PPV. Factors significantly associated with SPV perpetration were younger age (0.94 [0.91–0.98]), not being the household head (2.89 [1.01–8.28]), living together with the main partner (5.92 [2.14–16.38]), having a partner at least 5 years younger than themselves (2.62 [1.14–6.02]), reporting more than one partner in the 12 months prior to the survey (2.90 [1.57–5.37]), and having had lifetime sex with another man (8.07 [2.28–28.58]). Finally, factors significantly associated with SV perpetration included not being the household head (2.37 [1.02–5.50]), reporting inconsistent condom use with at least one of the 2 most recent partners (2.52 [1.21–2.26]), having more than 20 partners in one's lifetime (6.22 [1.76–21.97]) and perceivingg HIV-related stigma (1.21 [1.10–1.39] per unit).

**Table 1. Characteristics of HIV-positive men who declared at least one female partner in the 12 months prior to the survey[a] (ANRS-122988 EVOLCam survey, n = 406).**

| | N(%) or median [IQR] |
|---|---|
| *DEMOGRAPHIC AND SOCIOECONOMIC CHARACTERISTICS* | |
| **Age** (years) [b] | 43 [37;51] |
| **Living setting** | |
| Urban area | 337 (83) |
| Rural area | 69 (17) |
| **Educational level** | |
| No schooling or primary level | 99 (24) |
| Junior high school Secondary 1st cycle level | 132 (33) |
| High school | 106 (26) |
| Third level | 66 (16) |
| **Occupational category** | |
| Farmer | 57 (14) |
| Craftsperson, trader or business manager | 88 (22) |
| Managerial and professorial professional | 9 (2) |
| Intermediate professional | 37 (9) |
| Employee (civil service, commercial, service industry, etc.) or blue-collar worker | 128 (32) |
| Inactive | 87 (21) |
| **Monthly household income per** adult-equivalent (FCFA) | 11173 [5310;21429] |
| (USD) | 19 [9;36] |
| *DOMESTIC AND DYADIC CHARACTERISTICS* | |
| **Household head** | 361 (89) |
| **Number of children** | |
| None | 27 (7) |
| 1–4 children | 260 (64) |
| ≥5 children | 118 (29) |
| **Having a female main partner** | |
| Yes | 359 (88) |
| No | 47 (12) |
| Separated from partner because of HIV in the 12 months prior to the survey | 7 (2) |
| **Living with main partner** | 294 (72) |
| **Type of union** | |
| Legal or customary marriage | 210 (52) |
| Common-low union | 141 (35) |
| **Polygamous union** | 54 (13) |
| **Length of relationship with main partner** | |
| Not concerned (no main partner) | 47 (12) |
| < 4 y | 101 (25) |
| 5–8 y | 82 (20) |
| 9–16 y | 80 (20) |
| <16 y | 87 (21) |
| **In relationship with same main partner at the time of positive HIV diagnosis** | 164 (40) |
| **Currently desiring or trying to have a child with main partner** | 237 (58) |
| **Partner had higher educational level than respondent** | 68 (17) |
| **Partner was younger than respondent** (>5 years) | 307 (76) |
| **Main partner involved in decision-making about how to spend respondent's income** [c] | 154 (38) |
| **Main partner involved in decision-making about respondent's healthcare** [c] | 123 (30) |

(*Continued*)

**Table 1.** (Continued)

| | N(%) or median [IQR] |
|---|---|
| **Serodiscordant couple** (main partner HIV-negative or of unknown status) | 215 (53) |
| *PERPETRATED IPV* | |
| **Psychological and physical violence (PPV)** | |
| Moderate level [d] | 58 (14) |
| High level [e] | 55 (14) |
| **Severe physical violence (SPV)** | |
| Moderate level [d] | 27 (7) |
| High level [e] | 34 (8) |
| **Sexual violence (SV)** | |
| Moderate level [d] | 24 (6) |
| High level [e] | 22 (5) |
| *SEXUAL BEHAVIORS AND CHARACTERISTICS OF PARTNER(S)* [f] | |
| **≥Two partners in the 12 months prior to the survey** | 127 (31) |
| **Inconsistent condom use** with at least one of the 2 most recent (i.e., prior 12 months) female partners | 226 (56) |
| At least one of the 2 most recent partners was **HIV negative or of unknown status** | 285 (70) |
| **Positive HIV status disclosure to partner(s)** | |
| Not disclosed to at least one of the 2 most recent partners | 129 (32) |
| Not disclosed to at least one of the 2 most recent HIV negative or unknown status partners | 128 (32) |
| **Transactional sex in the 12 months prior to the survey** | |
| Bought sex | 16 (4) |
| Sold sex | 2 (0) |
| No transactional sex | 388 (96) |
| **Number of female partners in lifetime** | |
| 1–5 | 92 (23) |
| 6–10 | 82 (20) |
| 11–20 | 76 (19) |
| 21–50 | 46 (11) |
| >50 | 60 (15) |
| Unknown | 43 (11) |
| **Had sex with another man in lifetime** | 10 (3) |
| *CLINICAL CHARACTERISTICS AND EXPERIENCE OF LIVING WITH HIV* | |
| **Time since diagnosis** (years, months) | 3y9m [1y5m;6y10m] |
| **HIV diagnosis following diagnosis of partner (current or former)** | 164 (40) |
| **On ART** | |
| Yes, ≥6 months | 318 (78) |
| Yes, < 6 months | 41 (10) |
| No | 47 (12) |
| **Time between diagnosis and ART initiation** | |
| <1 month | 139 (34) |
| 2–3 months | 51 (13) |
| 4–15 months | 80 (20) |
| >15 months | 89 (22) |
| Not concerned (not receiving ART) | 47 (12) |
| **ART adherence in the 4 weeks prior to the survey** | |
| ART interruption≥ 2days | 84 (21) |

(*Continued*)

**Table 1.** (Continued)

| | N(%) or median [IQR] |
|---|---|
| Poor | 171 (42) |
| High | 104 (26) |
| Not concerned (not on ART) | 47 (12) |
| **ART interruption ≥1 month since ART initiation** | |
| No | 305 (75) |
| Yes, in the 6 months prior to the survey | 21 (5) |
| Yes, more than 6 months prior to the survey | 30 (7) |
| Not concerned (not on ART) | 47 (12) |
| **Belief in ART as effective tool for preventing sexual and mother-to-child HIV transmission** [g] | 106 (26) |
| **HIV-related stigma score** (range 0–8) | 0 [0;1] |
| PSYCHOSOCIAL CHARACTERISTICS | |
| **Mental health quality of life score** (range 0–100) | 47 [39;54] |
| **Frequent binge drinking** [b] | 51 (23) |

ART: Antiretroviral therapy; IQR: InterQuartile range.

[a] no missing data in the IPV questionnaire.

[b] in four categories defined using quartiles.

[c] replied "My partner" or "My partner and I together" vs. "Me alone", "Me and someone else", or "Someone else" to the questions "Who usually decides how you spend your income?" and "Who has the final say regarding decisions about your healthcare?"

[d] IPV score < median IPV score.

[e] IPV score ≥median IPV score.

[f] most recent partner if only one declared partner in the 12 months prior to the survey, or the 2 most recent partners if more than one partner declared in the 12 months prior to the survey.

[g] replied "a lot" or "quite a lot" to the question "How much do you think ART reduces the risk of HIV transmission during sex?" and to the question "How much do you think ART reduces the risk of HIV transmission to a baby during pregnancy?" (vs. "Do not know", "not at all", "a little" to at least one of the questions).

## Discussion

Our study highlighted that the three forms of IPV which we constructed were common in MLHIV linked to care in the national ART access program in the Littoral and Center regions of Cameroon. More specifically, PPV, SPV and SV, were reported by 28, 15 and 11% of the study population, respectively. Our findings also suggest that MLVIH reporting IPV had a higher risk of transmitting HIV, defined as a combination of unstable aviremia and inconsistent condom use. Additionally, our study highlighted that HIV-risk sexual behaviors and socioeconomic vulnerability played an important role in all three forms of IPV while perceived HIV-related stigma was more specifically associated with PPV and SV.

### IPV perpetration and factors related to sexual behaviors

Several behavioral risk factors for HIV transmission—including inconsistent condom use, lifetime sex with another man and having more than one partner—were associated with the three distinct forms of IPV identified in our study (PPV, SPV and SV). This finding is in line with previous research in Sub-Saharan Africa [40–42]. More precisely, men perpetrating SV were more likely to inconsistently use condoms, which in turn increased HIV transmission risk.

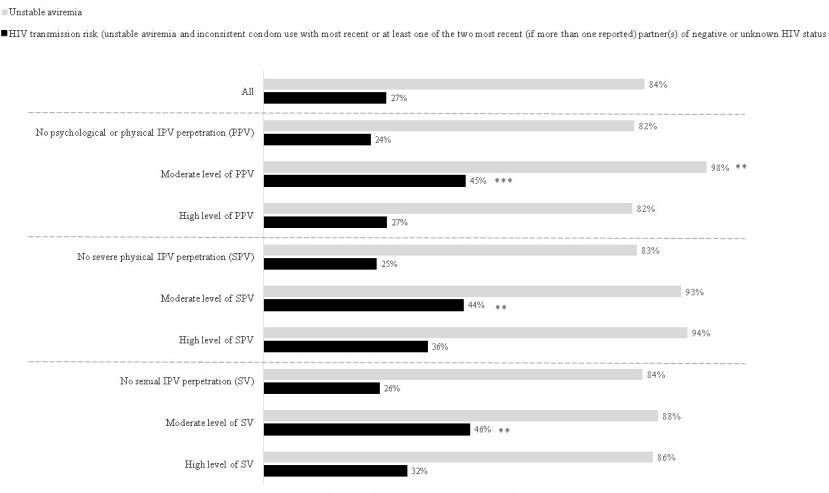

**Fig 1. Unstable aviremia and HIV transmission risk according to IPV perpetration (n = 406).**

Several studies have shown that men perpetrating sexual and/or physical violence may use various condom use resistance behaviors—including pressure, lies about their STI status, threats and persuasion—to put their partners in a situation where they cannot negotiate condom use or refuse condomless sex [7, 9, 43]. Furthermore, in our study, PPV and SPV perpetrators were more likely to have had more than one partner in the 12 months prior to the EVOLCam survey. Having multiple partners is a risky behavior for HIV transmission commonly associated with various forms of IPV perpetration [40–42, 44, 45]. In our study, an additional factor associated with perpetrating SPV included lifetime sex with other men. A study conducted in several West African countries among men who have sex with both men and women showed that those with lower self-acceptance of homosexuality were more likely to perpetrate IPV against women, possibly because heterosexual relationships are driven by social pressure, and that this violence was in turn associated with inconsistent condom use with female partners [46]. In addition, SV perpetrators in our study were more likely to report inconsistent condom use and more than 20 partners in their lifetime. Our findings suggest that SV perpetration and associated risky HIV sexual behaviors could generate disproportionate exposure to HIV infection for women, who are generally more monogamous than men, as observed in women participating in the EVOLCam survey [47] and elsewhere in Sub-Saharan Africa [40]. This is also clearly illustrated by national HIV incidence data which showed that women represent two thirds of new HIV infections in adults [22].

## IPV perpetration and living with HIV

Perceiving HIV-related stigma was significantly associated with both SV and PPV. Although this association has, to our knowledge, not been previously highlighted in MLVIH, in another study conducted in women living with HIV (WLHIV) using the EVOLCam data, perceived HIV-related stigma was associated with IPV victimization [47]. Suffering from social rejection or isolation due to positive HIV status probably impairs the mental quality of life of MLHIV, which is a predictor for perpetrating IPV [48, 49]. In our study, the three forms of IPV were either significantly associated or tended to be associated (at the 10% level) with a lower mental quality of life score in univariate analysis, something which may possibly mediate the relationship between IPV perpetration and perceiving HIV-related stigma.

**Table 2. Factors associated with IPV perpetration (n = 406)—Results from univariate ordinal logistic regressions.**

| | Psychological and physical IPV (PPV) | Severe physical IPV (SPV) | Sexual IPV (SV) |
|---|---|---|---|
| | Univariate analysis | Univariate analysis | Univariate analysis |
| | OR [95% CI] | OR [95% CI] | OR [95% CI] |
| *DEMOGRAPHIC AND SOCIOECONOMIC CHARACTERISTICS* | | | |
| **Age** (years) | 0.97 [0.95;0.95]*** | 0.94 [0.91;0.91] | 0.97 [0.94;0.94]** |
| **Living in a rural area** (ref. urban) | 1.73 [0.91;0.91]** | 2.16 [1.08;1.08]*** | |
| **Occupational category** (ref. farmer) | | | |
| Craftsperson, trader or business manager shopkeepers | | | 0.23 [0.06;0.06]*** |
| Managerial and professorial profession | | | |
| Intermediate professional | | | |
| Employee (civil service, commercial, service industry, etc.) or blue-collar worker | | | |
| Inactive | 1.87 [0.88;0.88]* | | |
| **Monthly household income** (1000 FCFA per adult-equivalent) (USD) | 0.98 [0.96;0.96]**** | 0.98 [0.97;0.97]* | |
| *DOMESTIC AND DYADIC CHARACTERISTICS* | | | |
| **Not the household head** (ref. being) | 1.44 [0.76;0.76] | 1.87 [0.88;0.88]* | 2.16 [0.96;0.96]** |
| **Having a main partner** (ref. not having one) | 1.83 [0.85;0.85]* | 2.79 [0.84;0.84]** | |
| **Living together with main partner** (ref. not living) with) | 1.43 [0.86;0.86]* | 2.34 [1.11;1.11]*** | |
| **Length of relationship with main partner** (ref. <4 y) | | | |
| Not concerned (no main partner) | | | |
| 5–8 y | | | |
| 9–16 y | | | 2.41 [0.95;0.95]** |
| <16 y | | | |
| **In relationship with main partner at time of positive HIV diagnosis** (ref. no) | 0.71 [0.45;0.45]* | | |
| **Currently desiring or trying to have a child with main partner** | 1.5 [0.96;0.96]** | | |
| **Partner younger than respondent** (>5 years) (ref. older or same age) | | 1.79 [0.86;0.86]* | |
| **Partner had higher educational level** than respondent (ref. lower or equal) | 0.66 [0.35;0.35]* | | 0.39 [0.13;0.13]** |
| **Main partner involved in decision-making about how to spend respondent's income** [a] (ref. not involved) | | 0.7 [0.4;0.4]* | |
| *SEXUAL BEHAVIORS AND PSYCHOSOCIAL CHARACTERISTICS* | | | |
| **≥Two partners in the 12 months prior to the survey** (ref. one) | 1.92 [1.23;1.23]**** | 2.29 [1.32;1.32]**** | 1.8 [0.96;0.96]** |
| **Inconsistent condom use with at least one of the 2 most recent partners** (ref. consistent condom use) | 1.46 [0.94;0.94]** | 2.06 [1.15;1.15]*** | 2.79 [1.37;1.37]**** |
| **Number of female partners in lifetime** (ref.1-5) | | | |
| 6–20 | | | |
| >20 | 1.73 [0.92;0.92]** | 2.25 [0.97;0.97]** | 7.21 [2.07;2.07]**** |
| Unknown | 1.72 [0.78;0.78]* | | |
| **Sex with another man in lifetime** (ref. never) | 3.47 [1.07;1.07]*** | 9.27 [2.78;2.78]**** | 4.14 [1.06;1.06]*** |
| **Mental quality of life** (per unit) | 0.98 [0.96;0.96]** | 0.97 [0.95;0.95]*** | 0.96 [0.94;0.94]*** |
| **Frequent binge drinking** [b] (ref. < once a month) | 2.19 [1.22;1.22]**** | 2.2 [1.1;1.1]*** | 2.21 [1.03;1.03]*** |
| *CLINICAL CHARACTERISTICS AND EXPERIENCE OF LIVING WITH HIV* | | | |
| **Time since diagnosis** (years, months) | | | 0.94 [0.86;0.86]* |
| **Time between diagnosis and ART initiation** (ref. <1 month) | | | |
| Not receiving ART | | | |

*(Continued)*

**Table 2.** (Continued)

| | Psychological and physical IPV (PPV) | Severe physical IPV (SPV) | Sexual IPV (SV) |
|---|---|---|---|
| 2–3 months | 1.91[0.98;0.98]** | | |
| 4–15 months | | 2.08 [1;1]** | |
| >15 months | | | |
| **ART adherence in the 4 weeks prior to the survey** (ref. high adherence) | | | |
| Poor adherence | 1.66[0.85;0.85]* | 2.33 [1;1]** | |
| ART interruption≥ 2days | 1.9[1.06;1.06]*** | 1.75 [0.81;0.81]* | |
| Not receiving ART | 2[0.92;0.92]** | 1.98 [0.73;0.73]* | |
| **Belief in ART as an effective tool in preventing sexual and mother-to-child HIV transmission** [c] (ref. does not believe it, or does not know) | 0.66[0.42;0.42]** | | |
| **HIV-related stigma score (per unit)** | 1.15[1.04;1.04]**** | 1.12 [0.99;0.99]** | 1.26 [1.12;1.12]**** |

OR: odds ratio; aOR: adjusted odds-ratio; CI: confidence interval; p-value [0.1;0.2 [*, [0.05;0.1[**, [0.01;0.05[***, <0.01 ****

[a] replied "My partner" or "My partner and I together" to the question "Who usually decides how you spend your income?" (ref. "Me alone" "Me and someone else" "Someone else")

[b] drinking ≥3 big bottles of beer (i.e., ≥ 260 cL total) or 6 other alcohol drinks on one occasion (once a month, a week, a day; less than once a month),

[c] replied "a lot" or "quite a lot" to the question "How much do you think ART reduces the risk of HIV transmission during sex?" and to the question "How much do you think ART reduces the risk of HIV transmission to a baby during pregnancy?" (vs. "Do not know", "not at all", "a little", to at least one of these questions)

NB: The following variables are not presented in this table either because they had a p-value >0.2 in the univariate analysis for the three IPV outcomes, or because only the category 'not concerned' (i.e., no main partner) had a p-value<0.20 in the univariate analysis. Consequently the following variables were not introduced in the multivariate analysis: educational level, number of children, polygamous union, type of union, main partner involved in decision-making about respondent's healthcare, serodiscordant couple (main partner's HIV status negative or unknown), HIV status of the 2 most recent partners, HIV status disclosure to at least one of the 2 most recent partners, transactional sex, separated from main partner because of HIV, on ART in the 4 weeks prior to the survey, ART interruption ≥1 month since ART initiation, HIV diagnosis following partner's (current or former) diagnosis.

The internalized construction of masculinity in Cameroon might also prevent men from disclosing their positive status and seeking help from relatives or healthcare [50–52]. Living with HIV may disrupt the normative male identity [44–46]. As suggested by our findings, additional psychosocial issues may arise in MLHIV who feel stigmatized and rejected because of their HIV infection, increasing the risk of IPV perpetration.

## IPV perpetration and MLHIV/partner's socioeconomic vulnerability

Our study also identified several demographic, socioeconomic and dyadic risk factors for IPV perpetration, which were not directly related to living with HIV. First, younger age was associated with both increased PPV and SPV, reflecting findings elsewhere both in PLHIV [53] and in the general population [48]. Violence against female partners is internalized by both young men and women in Cameroon [54] and in other Sub-Saharan settings [52]. Aggressiveness and violence may be used by young men to compensate for their lack of financial power, when competing with older and wealthier men [52]. Our findings also showed that MLHIV with a low household income and those who were not the household head were more likely to perpetrate SPV and SV, respectively. These findings suggest that socioeconomic vulnerability increases the risk of perpetrating IPV in MLVIH. Interestingly, studies investigating IPV victimization did not identify socioeconomic risk factors in women living with HIV (WLHIV) (using the EVOLCam data) or in the general Cameroonian female population [23, 47]. This suggests that socioeconomic vulnerability may play a role in IPV perpetration but not in IPV victimization, maybe because all socioeconomic categories of women are potentially affected.

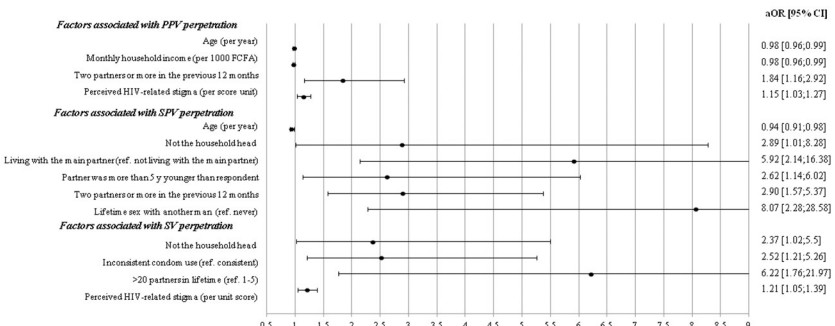

**Fig 2. Factors significantly associated with IPV perpetration in multivariate analysis.**

In addition, we found a significant effect of age disparity in partnerships (respondents being older) on SPV perpetration, which is consistent with a study in Tanzania where age disparity was associated with increased physical IPV victimization of women independently of their HIV status [55], probably because of unequal power dynamics. Cohabitation, a recognized domination factor for increased IPV [56], was also associated with SPV perpetration in our study, and with physical IPV victimization in the study conducted in WLHIV using the EVOLCam data [47].

## IPV perpetration and HIV transmission

Our findings show that almost half of the MLHIV who reported moderate PPV, SPV and SV had a risk of transmitting HIV (defined here as a combination of unstable aviremia and inconsistent condom use). In addition, IPV perpetrators were more likely to have lifetime and recent HIV-risky sexual behaviors. Those two findings suggest that IPV perpetrators had a high risk of transmitting HIV to their female partners. However, evidence is mixed in the literature regarding the relationships between HIV acquisition in women and experiencing various forms of IPV victimization. One study using national Demographic and Health surveys conducted in 10 Sub-Saharan countries showed that being HIV-infected was strongly associated with physical violence victimization but only in settings with high HIV prevalence (>5%) [3]. Indeed in the two regions of Cameroon which we studied here, the HIV epidemic is generalized with prevalence rates ranging from approximately 4% to 6%. However, in a meta-analysis performed on data from 10 developing countries, no association between IPV victimization and HIV infection was found, perhaps because the data came from countries with very different local HIV epidemics, ranging from hyper-endemic to highly concentrated HIV clusters [2]. In addition, a model-based study using South African data suggested that the association between IPV and HIV acquisition in women is likely due to confounding behavioral factors, especially perpetrators having multiple partners [57]. To our knowledge, no study to date has investigated the role of ART adherence and viral suppression in HIV transmission among IPV perpetrators. Our study suggests that HIV-viremia is more frequent among PPV perpetrators, which could be due to poor adherence to ART or treatment interruption, as observed in IPV victims [47, 58]. However, although we found that PPV perpetrators were more likely to report ART interruptions, the association was only significant in the univariate analysis. Given the difficulties faced by MLVIH perpetrating IPV, and deeply internalized masculinity norms which possibly prevent them for seeking healthcare [59, 60], further research examining the different steps of the HIV cascade of care among this population is needed.

## Strengths and limitations

Our analysis of factors associated with IPV perpetration was comprehensive, covering a large number of variables of different types (demographic, socioeconomic, domestic, dyadic, behavioral and psychosocial). To our knowledge, this is the first study to investigate IPV perpetrated by Cameroonian MLHIV, associated individual and dyadic factors, and its relationship with HIV transmission risk.

Despite these strengths, several limitations must be recognized. Our study's cross-sectional design did not allow us to infer any causality between IPV and the studied covariates. Longitudinal studies would be useful to improve knowledge about interactions between IPV perpetration and HIV acquisition and transmission, as well as specific issues affecting MLHIV. In addition, some information which may have influenced IPV perpetration was lacking, including post-traumatic stress, experience of family-of-origin violence, and witnessing inter-parental violence during childhood [6, 61]. We cannot therefore completely exclude the risk of bias in the estimations of the parameters of the multivariate model due to the omission of these potential relevant covariates.

Furthermore, our study relied on self-reports of IPV perpetration, possibly inducing declarative bias due to strong social desirability bias [62]. Compared with SV victimization declared by WLHIV in the EVOLCam survey [47], SV perpetration declared by MLHIV was lower in the present analysis. This disparity was also observed at the national level in Cameroon for SV [23] and in other studies where perpetrators and victims of sexual and/or physical IPV were selected from the same population [8, 41]. However, as observed by Campbell et al. [63], well-validated assessment protocols to identify IPV perpetrators are lacking. We therefore collected IPV perpetration data by adapting the WHO questionnaire designed and validated for IPV victimization, which is considered to be highly reliable in discriminating various forms of IPV against women in different settings [64]. In addition, we performed a principal component analysis to construct IPV scores from a large number (12) of IPV-specific questionnaire items. This method was also used in another study as it has the advantage of being able to separately examine the different dimensions of IPV and evaluate the level of each [65].

## Implications for public health policy

In Cameroon, where HIV and IPV are endemic and interrelated, screening for and preventing IPV perpetration should be included in counseling for male patients, with the wider goal of preventing HIV transmission to their partners. Beyond individual and dyadic characteristics, the effect of social masculinity construction and gender norms on men's attitudes and behaviors—including IPV perpetration—in the context of the HIV epidemic, should also be investigated at the community and societal levels [50].

Several HIV prevention interventions targeting women focus on empowering them to refuse condomless sex and not to accept their partner's infidelity. However, such interventions might increase their exposure to IPV [40]. Successful interventions to prevent IPV in Sub-Saharan Africa often incorporate HIV prevention and focus more on community building and engagement—targeting men in particular—than on individual approaches. More specifically, these interventions include using the support of community leaders and encouraging people to work together on various social dimensions including stereotypes, behaviors, gender-related issues of violence and sexuality, health consequences of IPV, and acceptability by women of their right to refuse to have sex [40]. We also suggest that including the detection and prevention of IPV perpetration in HIV counseling is an opportunity to reduce both IPV and HIV transmission.

## Conclusion

In our study, 28, 15 and 11% of participating MLHIV reported being perpetrators of PPV, SPV and SV, respectively. Socioeconomic vulnerability of both MLHIV participants and their female partners, as well as difficulties associated with living with HIV were associated with a higher likelihood of IPV perpetration. IPV perpetrators were also more likely to have lifetime and recent HIV-risky behaviors, which suggests an increased risk of transmitting HIV to their female partners. HIV research should further investigate the relationship between IPV perpetration and HIV transmission risk.

## Supporting information

**S1 File. Questionnaire modules in French.**
(PDF)

**S2 File. Questionnaire modules in English.**
(PDF)

**S1 Table. Frequency of 12 perpetrated IPV questionnaire items and detailed results of principal components analysis (PCA) used to construct IPV scores for each form of IPV from these questionnaire items.**
(DOCX)

## Acknowledgments

We thank all the participants and all the staff from the study's 19 HIV centers who agreed to take part in the EVOLCam survey. We also thank Gwenaëlle Maradan for the monitoring of data collection and Jude Sweeney for revising and editing the English version of the manuscript.

The EVOLCam study group: C. Kuaban, L. Vidal (principal investigators); G. Maradan, A. Ambani, O. Ndalle, P. Momo, C. Tong (field coordination team); S. Boyer, L. March, M. Mora, L. SagaonTeyssier, M. de Sèze, B. Spire, M. Suzan-Monti (UMR912 –SESTIM); C. Laurent, F. Liégeois, E. Delaporte, V. Boyer, S. Eymard Duvernay (TransVIHMI); F. Chabrol, E. Kouakam, O. Ossanga, H. Essama Owona, C. Biloa, M.T. Mengue (UCAC); E. MpoudiNgolé (CREMER); P.J. Fouda, C. Kouanfack, H. Abessolo, N. Noumssi, M. Defo, H. Meli (Hôpital Central, Yaoundé); Z. Nanga, Y. Perfura, M.Ngo Tonye, O. Kouambo, U. Olinga, E Soh (Hôpital Jamot, Yaoundé); C. Ejangue, E. Njom Nlend, A. Simo Ndongo (Hôpital de la Caisse, Yaoundé); E Abeng Mbozo'o, M. Mpoudi Ngole, N. Manga, C. Danwe, L. Ayangma, B. Taman (Hôpital Militaire, Yaoundé); E.C. Njitoyap Ndam, B. Fangam Molu, J. Meli, H. Hadja, J. Lindou (Hôpital Général, Yaoundé); J.M. Bob Oyono, S. Beke (Hôpital Djoungolo, Yaoundé); D. Eloundou, G. Touko, (District Hospital, Sa'a); J.J. Ze, M. Fokoua, L.Ngum, C.Ewolo, C.Bondze (District Hospital, Obala); J.D. Ngan Bilong, D. S.Maninzou, A. Nono Toche (Hôpital St Luc, Mbalmayo); M.Tsoungi Akoa, P. Ateba, S. Abia (District Hospital, Mbalmayo); A. Guterrez, R. Garcia, P. Thumerel (Catholic Health Centre, Bikop); E. Belley Priso, Y Mapoure, A. Malongue, A.P. Meledie Ndjong, B. Mbatchou, J. Hachu, S. Ngwane (Hôpital Général, Douala); J. Dissongo, M. Mbangue, Ida Penda, H. Mossi, G. Tchatchoua, Yoyo Ngongang, C.Nouboue, I. Wandji, L. Ndalle, J. Djene (Hôpital Laquintinie, Douala); M.J. Gomez, A. Mafuta, M. Mgantcha (Catholic Hospital St Albert Legrand, Douala); E.H. Moby, M.C. Kuitcheu, A.L. Mawe, Ngam Engonwei (District Hospital, Bonassama); L.J. Bitang, M. Ndam, R.B.Pallawo, Issiakou Adamou, G.Temgoua (District Hospital,Deido); C.Ndjie Essaga, C. Tchimou, A. Yeffou, I. Ngo, H. Fokam, H. Nyemb (District Hospital, Nylon); L.R. Njock, S. Omgnesseck, E.

Kamto, B. Takou (District Hospital, Edea); L.JG Buffeteau, F. Ndoumbe, JD Noah, I. Seyep (Hôpital St Jean de Malte, Njombe).

Lead author of the EVOLCam study group: Sylvie BOYER (email: sylvie.boyer@inserm.fr).

## Author Contributions

**Conceptualization:** Marion Fiorentino, Marie-Thérèse Mengue, Laurent Vidal, Christopher Kuaban, Laura March, Christian Laurent, Bruno Spire, Sylvie Boyer.

**Formal analysis:** Marion Fiorentino, Abdourahmane Sow, Luis Sagaon-Teyssier.

**Investigation:** Marie-Thérèse Mengue, Laurent Vidal, Christopher Kuaban, Laura March, Christian Laurent, Sylvie Boyer.

**Methodology:** Marion Mora, Laurent Vidal, Laura March, Bruno Spire.

**Project administration:** Marion Mora.

**Supervision:** Marion Fiorentino, Christopher Kuaban.

**Validation:** Marie-Thérèse Mengue.

**Writing – original draft:** Marion Fiorentino.

**Writing – review & editing:** Marion Fiorentino, Christian Laurent, Bruno Spire, Sylvie Boyer.

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
