## [Decision Letter · Decision Letter 0]

11 Dec 2020

PONE-D-20-36285

Intimate partner violence by men living with HIV in Cameroon: prevalence, associated factors and implications for HIV transmission risk (ANRS-12288 EVOLCAM)

PLOS ONE

Dear Dr. Fiorentino,

Thank you for submitting your manuscript to PLOS ONE. After careful consideration, we feel that it has merit but does not fully meet PLOS ONE’s publication criteria as it currently stands. Therefore, we invite you to submit a revised version of the manuscript that addresses the points raised during the review process.

We look forward to receiving your revised manuscript.

Kind regards,

Marianna Mazza

Academic Editor

PLOS ONE

Journal Requirements:

2. Please include additional information regarding the survey or questionnaire used in the study and ensure that you have provided sufficient details that others could replicate the analyses. For instance, if you developed a questionnaire as part of this study and it is not under a copyright more restrictive than CC-BY, please include a copy, in both the original language and English, as Supporting Information, or include a citation if it has been published previously.

3. In the Methods, please discuss whether and how the questionnaire was pre-tested. If these did not occur, please provide the rationale for not doing so.

5. One of the noted authors is a group or consortium [EVOLCam study Group]. In addition to naming the author group and listing the individual authors and affiliations within this group in the acknowledgments section of your manuscript, please also indicate clearly a lead author for this group along with a contact email address.

Reviewers' comments:

Reviewer's Responses to Questions

**Comments to the Author**

1. Is the manuscript technically sound, and do the data support the conclusions?

Reviewer #1: Yes

2. Has the statistical analysis been performed appropriately and rigorously? 

Reviewer #1: Yes

3. Have the authors made all data underlying the findings in their manuscript fully available?

Reviewer #1: Yes

4. Is the manuscript presented in an intelligible fashion and written in standard English?

Reviewer #1: Yes

5. Review Comments to the Author

Reviewer #1: The manuscript describes the relationship between intimate partner violence and men living with HIV in Cameroon. The theme is adequately explored and with a flowing form. It could be stimulating a reflection on the victimization process and the connected psychopathological configurations, as well as on the fact that challenging circumstances and resilience skills can contribute to a better coping with stress and adversity.

It might be interesting to analyze the impact of COVID19 pandemic on emotional distress and sexuality. It would therefore be useful to read and discuss the following papers:

G Marano, E Gaetani, A Gasbarrini, L Janiri, G Sani, M Mazza, Multidisciplinary Gemelli Group for HHT. Mental health and counseling intervention for hereditary hemorrhagic telangiectasia (HHT) during the COVID-19 pandemic: perspectives from Italy. Eur Rev Med Pharmacol Sci. 2020 Oct;24(19):10225-10227. doi: 10.26355/eurrev_202010_23246.

M Mazza, G Marano, L Janiri, G Sani. Managing Bipolar Disorder patients during COVID-19 outbreak. Bipolar Disord. 2020 Oct 4. doi: 10.1111/bdi.13015.

M Mazza, G Marano, B Antonazzo, E Cavarretta, M Di Nicola, L Janiri, G Sani, G Frati, E Romagnoli. What about heart and mind in the covid-19 era? Minerva Cardioangiol. 2020 May 12. doi: 10.23736/S0026-4725.20.05309-8.

M Mazza, G Marano, C Lai, L Janiri, G Sani. Danger in danger: Interpersonal violence during COVID-19 quarantine. Psychiatry Res. 2020 Jul;289:113046. doi: 10.1016/j.psychres.2020.113046.

6. PLOS authors have the option to publish the peer review history of their article (what does this mean?). If published, this will include your full peer review and any attached files.

Reviewer #1: No

---

## [Author Response · Author response to Decision Letter 0]

8 Jan 2021

Marion FIORENTINO 

INSERM, IRD, Aix Marseille Univ

SESSTIM

marion.fiorentino@inserm.fr

PLOS One 

16h December 2020

Dear Editor,

We would like to thank you and the reviewer for the relevant comments. We addressed all the issues in our point-by-point responses below in italic. 

Yours faithfully, 

Marion Fiorentino, on behalf of the authors

Journal Requirements:

 We have carefully checked that our manuscript meets PLOS ONE’s style requirements. We changed headings formatting and file names accordingly. 

2. Please include additional information regarding the survey or questionnaire used in the study and ensure that you have provided sufficient details that others could replicate the analyses. For instance, if you developed a questionnaire as part of this study and it is not under a copyright more restrictive than CC-BY, please include a copy, in both the original language and English, as Supporting Information, or include a citation if it has been published previously.

We included French and English questions corresponding to data presented in the present article as S1 and S2 files. We referred to S1 and S2 in the “study design and data collection” section:

 “The questionnaire modules corresponding to data presented in this article are included in supporting information S1 (French) and S2 (English).

We accordingly improved translation of questions about partner violence in the renamed S3 file. 

3. In the Methods, please discuss whether and how the questionnaire was pre-tested. If these did not occur, please provide the rationale for not doing so.

Yes, our questionnaire was pre-tested in six hospitals in October 2013. We added in the study design and data collection section “Before the study’s implementation, a pilot survey was conducted to test the questionnaires and data collection procedures in six urban and rural hospitals.”

Due to French law there are restrictions on publicly sharing the data of this study. French law requires that everyone who wishes to access cohorts data or clinical study data on humans must ask the French data protection authority, la Commission Nationale de l'Informatique et des Libertés (CNIL), for permission by filling a form which can be provided by Gwenaëlle Maradan (Observatoire Régional de la Santé PACA ie. The Regional Health Observatory PACA mail: gwenaelle.maradan@inserm.fr ). For further information, please see: https://www.cnil.fr/.

We included these information in our revised cover letter. 

5. One of the noted authors is a group or consortium [EVOLCam study Group]. In addition to naming the author group and listing the individual authors and affiliations within this group in the acknowledgments section of your manuscript, please also indicate clearly a lead author for this group along with a contact email address.

 We added Sylvie Boyer as the lead author of the EVOLCam study group in the Acknowledgements. 

 We apologized for this mistake. The data about women participating in the EVOLCam survey, has now been published in an other article. Thus, we replaced “data not shown” by the reference: Fiorentino M, Sagaon-Teyssier L, Ndiaye K, Suzan-Monti M, Mengue M-T, Vidal L, et al. Intimate partner violence against HIV-positive Cameroonian women: Prevalence, associated factors and relationship with antiretroviral therapy discontinuity—results from the ANRS-12288 EVOLCam survey. Women’s Health. 2019;15: 1745506519848546. 

 We added captions of supporting information at the end of the manuscript and updated citations accordingly. 

Reviewers' comments:

Reviewer's Responses to Questions

Comments to the Author

1. Is the manuscript technically sound, and do the data support the conclusions?

Reviewer #1: Yes

2. Has the statistical analysis been performed appropriately and rigorously? 

Reviewer #1: Yes

3. Have the authors made all data underlying the findings in their manuscript fully available?

Reviewer #1: Yes

4. Is the manuscript presented in an intelligible fashion and written in standard English?

Reviewer #1: Yes

5. Review Comments to the Author

Reviewer #1: The manuscript describes the relationship between intimate partner violence and men living with HIV in Cameroon. The theme is adequately explored and with a flowing form. It could be stimulating a reflection on the victimization process and the connected psychopathological configurations, as well as on the fact that challenging circumstances and resilience skills can contribute to a better coping with stress and adversity.

We thank the reviewer for her/his supporting comments. 

It might be interesting to analyze the impact of COVID19 pandemic on emotional distress and sexuality. It would therefore be useful to read and discuss the following papers:

This is indeed an interesting topic, that should be considered as a full research itself. Research about partner violence is especially important under the current circumstances, as quarantine appears to have exacerbated partner violence. The following papers are therefore of primary interest. 

G Marano, E Gaetani, A Gasbarrini, L Janiri, G Sani, M Mazza, Multidisciplinary Gemelli Group for HHT. Mental health and counseling intervention for hereditary hemorrhagic telangiectasia (HHT) during the COVID-19 pandemic: perspectives from Italy. Eur Rev Med Pharmacol Sci. 2020 Oct;24(19):10225-10227. doi: 10.26355/eurrev_202010_23246.

M Mazza, G Marano, L Janiri, G Sani. Managing Bipolar Disorder patients during COVID-19 outbreak. Bipolar Disord. 2020 Oct 4. doi: 10.1111/bdi.13015.

M Mazza, G Marano, B Antonazzo, E Cavarretta, M Di Nicola, L Janiri, G Sani, G Frati, E Romagnoli. What about heart and mind in the covid-19 era? Minerva Cardioangiol. 2020 May 12. doi: 10.23736/S0026-4725.20.05309-8.

M Mazza, G Marano, C Lai, L Janiri, G Sani. Danger in danger: Interpersonal violence during COVID-19 quarantine. Psychiatry Res. 2020 Jul;289:113046. doi: 10.1016/j.psychres.2020.113046.

6. PLOS authors have the option to publish the peer review history of their article (what does this mean?). If published, this will include your full peer review and any attached files.

Do you want your identity to be public for this peer review? For information about this choice, including consent withdrawal, please see our Privacy Policy.

Reviewer #1: No

---

## [Editor Report · Decision Letter 1]

15 Jan 2021

Intimate partner violence by men living with HIV in Cameroon: prevalence, associated factors and implications for HIV transmission risk (ANRS-12288 EVOLCAM)

PONE-D-20-36285R1

Dear Dr. Fiorentino,

We’re pleased to inform you that your manuscript has been judged scientifically suitable for publication and will be formally accepted for publication once it meets all outstanding technical requirements.

Kind regards,

Marianna Mazza

Academic Editor

PLOS ONE
---

## [Editor Report · Acceptance letter]

4 Feb 2021

PONE-D-20-36285R1 

Intimate partner violence by men living with HIV in Cameroon: prevalence, associated factors and implications for HIV transmission risk (ANRS-12288 EVOLCAM) 

Dear Dr. Fiorentino:

I'm pleased to inform you that your manuscript has been deemed suitable for publication in PLOS ONE. Congratulations! Your manuscript is now with our production department. 

Kind regards, 

on behalf of

Dr. Marianna Mazza 

Academic Editor

PLOS ONE